# Human Microbiome Mixture Analysis Using Weighted Quantile Sum Regression

**DOI:** 10.3390/ijerph20010094

**Published:** 2022-12-21

**Authors:** Shoshannah Eggers, Moira Bixby, Stefano Renzetti, Paul Curtin, Chris Gennings

**Affiliations:** 1Department of Environmental Medicine and Public Health, Icahn School of Medicine at Mount Sinai, 1 Gustave L. Levy Place, Box 1057, New York, NY 10029, USA; 2Department of Medical-Surgical Specialties, Radiological Sciences and Public Health, Università degli Studi di Brescia, Piazza del Mercato, 15, 25121 Brescia, Italy

**Keywords:** human microbiome, microbiome analysis, mixture analysis, weighted quantile sum regression

## Abstract

Studies of the health effects of the microbiome often measure overall associations by using diversity metrics, and individual taxa associations in separate analyses, but do not consider the correlated relationships between taxa in the microbiome. In this study, we applied random subset weighted quantile sum regression with repeated holdouts (WQS_RSRH_), a mixture method successfully applied to ‘omic data to account for relationships between many predictors, to processed amplicon sequencing data from the Human Microbiome Project. We simulated a binary variable associated with 20 operational taxonomic units (OTUs). WQS_RSRH_ was used to test for the association between the microbiome and the simulated variable, adjusted for sex, and sensitivity and specificity were calculated. The WQS_RSRH_ method was also compared to other standard methods for microbiome analysis. The method was further illustrated using real data from the Growth and Obesity Cohort in Chile to assess the association between the gut microbiome and body mass index. In the analysis with simulated data, WQS_RSRH_ predicted the correct directionality of association between the microbiome and the simulated variable, with an average sensitivity and specificity of 75% and 70%, respectively, in identifying the 20 associated OTUs. WQS_RSRH_ performed better than all other comparison methods. In the illustration analysis of the gut microbiome and obesity, the WQS_RSRH_ analysis identified an inverse association between body mass index and the gut microbe mixture, identifying Bacteroides, Clostridium, Prevotella, and Ruminococcus as important genera in the negative association. The application of WQS_RSRH_ to the microbiome allows for analysis of the mixture effect of all the taxa in the microbiome, while simultaneously identifying the most important to the mixture, and allowing for covariate adjustment. It outperformed other methods when using simulated data, and in analysis with real data found results consistent with other study findings.

## 1. Introduction

The human microbiome is increasingly recognized as an important component of human health. Studies show links between the composition and function of the human gut microbiome and many health outcomes, including inflammatory and autoimmune conditions, obesity, infection, and neurological outcomes [1,2,3,4]. Animal studies have shown prospective changes in the microbiome from different exposures, and changes in physiology and health status after changes to the microbiome. While some clinical trials have been done, many human microbiome studies have been observational.

In observational studies, the microbiome is typically characterized in a few key ways. The first is by measuring and comparing within individual diversity, or α-diversity. These measures, adopted from the field of ecology, measure the number of different taxa present, and the evenness of abundance among those taxa within a single sample [5,6,7]. That diversity level can then be compared across individuals. However, α-diversity cannot be directly translated to health status, thus its meaningful utility is limited. A second way to characterize the microbiome is by assessing community composition, or β-diversity. This is typically done by measuring the similarity or dissimilarity of the composition of one sample compared to another, using the number of different taxa [8], the abundance of each taxa [9], and sometimes the phylogenetic relationships between taxa [10]. Researchers can compare groups, for instance exposed and unexposed groups, to see if the samples within one group are more similar than samples across groups (i.e., controls are more similar to other controls than to the exposed group). We use these measures of diversity to try to gain an understanding of the effect of or on the microbiome as a whole, but changes in diversity can only indicate a general difference without indicating how specifically the microbiome is different, or what the important players within the microbiome are. 

To determine which specific taxa contribute to differences in diversity, researchers can also assess each taxa individually by measuring the amount of variability each one contributes, or by assessing trends in the presence/absence or abundance of individual taxa. While combinations of these strategies are often used, these scenarios must be adjusted for multiple comparisons across hundreds or thousands of taxa, which limit the ability to identify statistically significant associations. Furthermore, the microbiome is an ecosystem of bacterial communities with complex interactions and associations, and using these modeling strategies to assess them individually does not account for their intricate correlations. 

Interest in microbiome research has grown rapidly over the last fifteen years, yet the complexity of the data, e.g., zero inflation, variation across individuals, correlated taxa, etc., continues to be a challenge for researchers. There has been a push for new statistical methodologies, including machine learning methods, and new microbial data applications of existing statistical methods, in an effort to improve the accuracy of findings from microbiome analyses [11]. Some of these methods include random forest [12], negative binomials [13], and clustering [14], to name a few. Similarly, there has been a push in the field of environmental epidemiology to develop new strategies to model the health effects of multiple co-exposures to improve the accuracy of chemical exposure studies. Some of the newly developed methods allow for analysis of overall mixture effects, and indicate the importance of each chemical within that mixture. One such method is weighted quantile sum (WQS) regression [15]. WQS regression uses an empirically weighted index of many correlated chemical exposure measurements, and models the mixture effect of the whole index, while also providing weights for each component within the mixture to indicate the relative importance. WQS regression also allows for the inclusion of covariates, to reduce the effects of confounding. Applying this method to analysis of microbiome data allows for the evaluation of the overall mixture effect of the microbiome, and simultaneously identifies the most important individual taxa in the mixture while accounting for a correlated data structure.

The goal of this study is to demonstrate the novel application of WQS regression to assess covariate-adjusted associations between health exposures and microbiome sequencing abundance data as a mixture of potentially correlated bacterial taxa. Our analysis adjusted and combined WQS regression with random subset selection [16] and repeated holdout [17] (WQS_RSRH_) frameworks, and applied them to publicly available Human Microbiome Project (HMP) 16S amplicon sequencing data. We further demonstrate the utility of the method using data from the Growth and Obesity Cohort Study (GOCS) in Chile. 

The random subset extension of WQS is used in cases where the number of components in the WQS index is greater than the number of observations, and uses random subsets of components to calculate the WQS index. The repeated holdout extension of WQS allows for more robust estimates by using different observations in the training and validation sets of the data over multiple iterations of WQS analysis. We illustrate the utility of WQS_RSRH_ and estimate specificity (correctly determining OTUs were not associated with the outcome) and sensitivity (correctly identifying associated OTUs) of the method. This methodological application allows for more comprehensive investigation of the association between the gut microbiome and many health exposures and outcomes by assessing the microbiome as a mixture.

## 2. Methods

Figure 1 illustrates a simplified flow chart of the methods used for this study. All methods were performed in accordance with the relevant guidelines and regulations. This study is not considered human subjects research and is exempt from review by the Mount Sinai Institutional Review Board, as the data are de-identified and publicly available.

### 2.1. Data Source and Processing for Simulation

Data for the simulation analysis came from the HMP, version 1, which has been well described in previous literature [18,19,20]. We used 16S amplicon sequencing data, processed using QIIME [19,21]. HMP guidelines were followed in this analysis and publication.

We used data from each participant’s first stool sample (n = 210). As a data reduction step, we filtered out any operational taxonomic unit (OTU) that had 0 abundance in more than 90% of samples, resulting in a total of 868 OTUs. This data reduction step also ensures that there are enough participants with non-zero values to calculate tertiles above zero for the WQS_RSRH_ indices. Relative abundances were calculated to account for variations between individuals within the sample population. These data processing steps were performed in SAS v 9.4, R v 3.6.1, and RStudio v 1.2.5001, using the HMP16SDATA package [22].

### 2.2. Data Simulation Method

Twenty OTUs were chosen based on a literature review of bacterial species that have been linked to health-related variables such as body mass index (BMI) and smoking status (see Table 1). The number of OTUs were chosen for ease of the simulation step, and are not a reflection of the WQS_RSRH_ model’s power to detect associated components. These 20 OTUs were then randomly categorized to represent levels of association (strong, medium, and weak) with a simulated binary variable. To simulate the “test” variable, the intercept (β_0_) was set to −5, and the potency adjusted relative abundance (calculated as log2(log10(x + 1)/log10(max + 1)) for each of the 20 OTUs) was multiplied by a β coefficient of 8 for the strong group, 4 for the medium group, and 2 for the weak group. Potency adjustment was needed to standardize the simulated association across samples because the abundance of each OTU varied greatly, i.e., multiplying by 8 leads to a different scale of association for an abundance of 5 vs. an abundance of 30. Beta coefficient values were chosen for each level of association that would result in logit values across the range of OTUs between roughly −3.5 and 3.5. In prior attempts at simulation, larger coefficient values resulted in unlikely and extreme logit values (i.e., greater than 3.5). All other OTUs were not assigned an association with the test variable (i.e., assumed to be a value of 0). The resulting test variable was positive for 13% of the participants. We adjusted the simulation model for sex using a beta coefficient of −1 for females; i.e., males were the reference group. We also simulated a random “control” variable that was not assigned an association to any OTU, and used it as a negative control to compare the ability of each model to detect the intended simulated association.

### 2.3. Weighted Quantile Sum Regression Analysis with Random Subsets and Repeated Holdouts

WQS is a method applied to mixtures of variables (e.g., chemicals, or in this case OTUs) by which the total effect of a group of potentially correlated predictors is estimated through the derivation of an index, a weighted sum of the quantiled exposure variables [15]. The WQS index is calculated as $WQS=\sum w_{i}q_{i,j}$, where *WQS* is the mixture index, $q_{i,j}\;$is the quantiled variable for the *i*th exposure variable and *j*th subject and$\;w_{i}\;$is the weight corresponding to $q_{i}$. In WQS with Random Subsets (WQS_RS_), subsets of the variables in the mixture are randomly chosen and used to predict weights in order to maximize the association between the index and the outcome. Such subsets are computed numerous times (e.g., 1000 times) with adjustments for covariates. The average weights across the subsets sum to 1 and are used to compute the final WQS index for a given health outcome. For the calculation of the weighted index, effects can be constrained in the positive or negative direction, or weights can be calculated without constraining direction. The WQS index is then used in a generalized linear model (GLM), so that $g\left(\mu \right)\approx \;\alpha +\beta_{1}WQS+\delta Z$, where *g*() indicates a link function, *μ* is the sample mean, α is the intercept, $\beta_{1}$ is the effect parameter corresponding to the WQS index, and *Z* represents a set of covariates with corresponding effect estimates *δ* [16]. To increase generalizability, the weights are estimated and tested in randomly selected training (40% subjects) and validation (60% subjects) datasets. Although analysis can be constrained in the positive or negative direction for the weighted index calculation, estimates from the GLM in the validation dataset are not constrained. Therefore, GLM estimates can be in either direction regardless of constraint direction in the index calculation. WQS_RS_ with Repeated Holdouts (WQS_RSRH_) then repeats the WQS_RS_ process a specified number of times, with different observations in the training and validation datasets, and provides effect estimates and mixture weights for each repetition of the analysis. In each repetition of the analysis, the predictors with the largest weights within the WQS index contribute most to the estimated effect parameter. An equi-weight (1/the number of components in the index) cut-point is often used to determine which components within the mixture are most important, as it indicates if an individual weight is higher than if all components of the mixture were given equal weight. Across repeated holdouts, average effect estimates and average component weights are calculated for more robust estimates.

For this analysis, we looked at the association between the test variable and the gut microbiome, in order to demonstrate the application of WQS to microbiome data. Because of the large amount of zeros across the dataset, using quantiles in the WQS index was not ideal due to having so many ties with a 0 score. Instead, we ranked the relative abundance of each OTU to four levels, 0, 1, 2, or 3, where the observed 0s were scored as 0, and values above 0 were tertile scored. Due to the large number of variables (OTUs) in the index, the random subset variation of WQS was used in this analysis. To address generalizability we conducted 30 repeated holdout analyses where the distribution of weights were based on the 30 training sets (40%) and the distribution of the 30 estimates of $\beta_{1}$ was based on the 30 holdout validation sets (60%). The weights within each training set were based on 1000 random subsets of size 30 OTUs. To calculate the WQS index weights from the 1000 random subsets, three weighted averages were evaluated using different signal functions. The signal function gives additional emphasis to subsets with a larger association to the outcome, compared to those sets with negligible association. The three signal functions tested were: (i) the default in the gWQS R package, which weights each random subset based on the squared t statistic for the corresponding beta parameter; (ii) a more severe weighted average, which weights using exp(t); i.e., the absolute value of the t statistic exponentiated; and (iii) a less severe weighted average, using the absolute value of the t statistic. 

Sensitivity and specificity of this application were then calculated based on the WQS_RSRH_ index weights of the OTUs across the 30 repeated holdout sets. Both sensitivity and specificity were calculated over a range of cut points to guide cut point selection. Sensitivity was calculated as the proportion of the 20 selected OTUs that had weights exceeding the given cut point; specificity was calculated as the proportion of the remaining 848 OTUs that had weights below the cut point. We evaluated the impact of the signal function in the weighted averages of the WQS_RSRH_ indexes using analysis of variance for sensitivity and specificity for the 30 holdout datasets across the 3 signal functions and selected cut points. Cut points ranged between 0.0005 and 0.002, where 1/868 = 0.00115 (the equi-weighted cut point). The test for interaction was used as a goodness-of-fit test for the main effect ANOVA model. 

### 2.4. Comparison to Other Microbiome Analysis Methods

We also compared the WQS_RSRH_ method to more standard methods of microbiome analysis, using the same data set with the same test and control variables. We used the Vegan package [33] in R to calculate α-diversity using the Shannon index [6], and β-diversity distance using the Bray–Curtis dissimilarity index [9]. We performed two linear regressions with Shannon diversity as the outcome and the test variable, and the control variable as the primary exposure in each model, both adjusted for sex. The adonis2 function in Vegan was then used to perform two permutational analysis of variance (PERMANOVA) analyses, based on the Bray–Curtis index, with the same variables as the linear regression, using 9999 permutations. As a sensitivity analysis, we also conducted the same PERMANOVA analysis using the Aitchison distance [34]. Similarity percentage (SIMPER) analysis was then used, with 999 permutations, to determine which OTUs contributed 70% of the variance to the β-diversity differences between the levels of the simulated variable, and the random variable in a separate analysis. As an additional comparison, Random Forest analysis was conducted using the randomForest package [35], with 100 trees, to identify the OTUs most associated with the test and control variables. Separate models were run with the test and control variables as the response in each model, and the 868 OTUs and sex as predictors.

### 2.5. Data Source and Processing for Demonstration with Real Data

We further demonstrated the utility of the WQS_RSRH_ method by using it to examine the relationship between BMI and the gut microbiome using real (not simulated) data from a cohort of adolescent girls from Chile. The study design of GOCS in Chile has been previously described [36,37], The current study assesses a subset of 161 girls that contributed stool samples, BMI z-score for age and sex, calculated using the World Health Organization Anthro Survey Analysers, and complete covariate data. Covariate and outcome data collected at the stool sample collection clinic visit include BMI, age, and antibiotic use in the past six months (yes/no). Covariate data collected from survey at study baseline (around 10 years of age) included birth mode (vaginal/C-section), maternal education (high school or less versus more than high school), and number of days the girl was breastfed as an infant. 

Data for this project was obtained from the publicly available data in the Human Health Exposure Analysis Resource (HHEAR) Data Repository, which has been approved under Icahn School of Medicine at Mount Sinai IRB Protocol # 16-00947. HHEAR data use guidelines were followed in this analysis and publication.

The microbiome taxonomy was assigned as amplicon sequence variant (ASV)s, as previously described [36]. ASV data was then further processed by removing any ASVs with ambiguous taxonomy, and limiting to ASVs detected in at least 10% of subjects (ASV n = 500). The relative abundance of ASVs were calculated for each subject. 

### 2.6. WQS_RSRH_ Demonstration with Real Data

The WQS_RSRH_ method was applied to these data, with BMI category (normal vs. overweight/obese) as the outcome. The WHO BMI for sex- and age- z-score categories were categorized as between −2 to +1 as normal weight, and 2+ as overweight/obese [38]. The ASV relative abundances were scored into 3 groups such that 0 abundance was maintained at 0, and the remaining abundances were split as less than the median (1) or greater than or equal to the median (2). The WQS microbiome mixture was analyzed at the ASV level of taxonomy, thus the weights estimated in relation to the BMI category were per ASV. Before implementing the WQS_RSRH_, WQS_RS_ analysis, adjusted for covariates, without directional constraints was run to determine the directionality of the association between the microbiome mixture and BMI, and then run again with directional constraints to confirm the direction of the association. There were 2000 random subsets with 22 ASVs randomly selected to contribute to each of the random subsets. The WQS_RSRH_ analysis was run with 30 repeated holdouts with the same parameters (2000 random subsets with 22 ASVs per subset) and adjusted for covariates. All WQS analyses were trained on 40% of the subjects and validated on the remaining 60%. ASV weights were then summed by taxonomy into genus-level weights, calculated as the sum of all ASVs within the genus. A genus-level threshold was calculated as *1/c*, *c* being the number of genera found in the microbiome mixture. Weights above the 1/c threshold indicate that the genus was more impactful on the outcome (BMI) than under the assumption that all genera were equally weighted, such that all genera had the same impact on BMI. 

Because the WQS_RSRH_ selects to train and validate on 40% and 60% of the subjects, respectively, the random selection of categorical variables (outcome and covariates) in this split can select a subset that contains all of the same category (say 0 or 1) of one or more variables. In this case, the analysis will not run. To avoid this issue, we partitioned the data such that the training and validation splits across the repeated holdouts ensured the outcome in each analysis maintained variability by containing subjects with each categorical level.

### 2.7. Comparison Method Analysis with Real Data

We compared the WQS_RSRH_ method to more standard methods of microbiome analysis using the real data from GOCS as well. We used the phyloseq package in R to calculate α-diversity using the Shannon index [6,39], and β-diversity distance using the Bray–Curtis dissimilarity index [9]. We used linear regressions with Shannon diversity as the outcome and BMI category as the primary exposure variable, adjusted for maternal education, birth mode, age, duration of breastfeeding, and antibiotic use. The adonis2 function in Vegan was used to perform PERMANOVA analysis, based on the Bray–Curtis index, with the same variables as the linear regression. SIMPER analysis was used to determine which OTUs contributed 70% of the variance to the β-diversity differences between the levels of BMI. Random Forest analysis was conducted using the randomForest package [35] to identify the OTUs most predictive of BMI category. 

## 3. Results

### 3.1. OTU Distribution

Table 1 shows a description of the 20 OTUs that were assigned an association with the test variable. The percentage of the 210 participants that had 0 abundance for each of the 20 OTUs ranged between 35.2% and 88.1%, with the *Staphylococcus* OTU having the most non-zero abundance. The OTU with the highest maximum relative abundance was the *Dorea* genus, with a relative abundance of 19.75% for one participant. 

### 3.2. WQS_RSRH_ Results

WQS_RSRH_ regression was conducted with an average of 13% positive in the test variable, and was adjusted for sex. Each model in the 30 repeated holdout sets used 1000 randomly selected sets of size 30 OTUs. The beta coefficient estimates in the 30 repeated holdout datasets were all positively associated with the test variable (Figure 2). In comparison, the WQS_RSRH_ index was not significantly associated with a random control variable in the same dataset (Figure 2). This test of association indicates that there is an association between the microbiome as a whole and the simulated test variable. 

The WQS weights indicate the importance of each individual OTU on the association between the simulated variable and the microbiome. Here, the maximum average weight across the repeated holdouts is 0.0107, roughly 10 times the size of the average weight, while the lowest quartile weight was 0.00036, roughly 1/3 of the average weight. 

### 3.3. WQS_RSRH_ Sensitivity and Specificity

A range of cutoff threshold values were evaluated for identifying OTUs associated with the probability of observing the binary outcome variable. Sensitivity (the proportion correctly identified with weights above the cutoff) and specificity (the proportion correctly not identified with weights less than the cutoff) were evaluated for each cutoff (Appendix A). The equi-weighted cutoff is 1/868 = 0.00115. The specificity is improved from the equi-weighted cutoff with a value of 0.00131 where both sensitivity and specificity are roughly 73%. Using the equi-weighted cutoff, average sensitivity is 75%, average specificity is 70%. The two OTUs modeled with a ‘strong’ association had average sensitivity of 87%; the 8 OTUs with a ‘medium’ association had an average sensitivity of 90%; and on average 61% of the 10 weak components were identified correctly (Table 2).

### 3.4. WQS_RSRH_ Signal Functions

We next evaluated potential differences in sensitivity and specificity using different signal functions; i.e., (i) the absolute value of the t statistic corresponding to the beta coefficient for WQS; (ii) the square of the t statistic; and (iii) exp(t). In the analysis of both the sensitivity and specificity estimates across the signal functions and cut points in ANOVA, the cross-product term was not significant, indicating an adequate fit for the main-effects ANOVA model. In reduced main-effect models, as anticipated, there was a significant improvement in specificity with more severe weighting: i.e., in increasing order of abs(t), t^2^, exp(t) (*p* < 0.001; Appendix A). However, there was no difference in sensitivity with changes in the signal function (*p* = 0.597; Appendix A).

### 3.5. Diversity Comparison

The average Shannon diversity (α-diversity) score was 4.14, ranging from 1.58–5.02. In a linear regression with Shannon diversity as the outcome, the test variable was associated with 0.14 increased score (*p* = 0.20), adjusted for sex (Male β = −0.06, *p* = 0.41). The same regression was performed with the random variable as the primary predictor, and found no association between the random variable and Shannon diversity (β = −0.03, *p* = 0.67). 

β-Diversity was calculated with the Bray–Curtis dissimilarity index (Figure 3). Using PERMANOVA, we assessed the association between the test variable, adjusted for sex, and β-diversity. We found no association with the test variable (R^2^ = 0.005, *p* = 0.35) or sex (R^2^ = 0.005, *p* = 0.35). We also found no association in the model with the control variable (R^2^ = 0.004, *p* = 0.71) and sex (R^2^ = 0.005, *p* = 0.47). In sensitivity analysis using the Aitchison distance instead of Bray–Curtis, results were similar with no association found with the test variable (R^2^ = 0.005, *p* = 0.38) or the control variable (R^2^ = 0.005, *p* = 0.54).

### 3.6. Comparison of OTU Identification

We used SIMPER analysis to identify the OTUs contributing 70% of the variance to the differences in composition (β-diversity) by level of the exposure variable. We ran separate analyses using the test and control variables as the exposure variable, obtaining very similar results. Both analyses identified 1 of 2 OTUs assigned a strong association, 5 of 8 OTUs assigned a medium association, and 2 of 10 OTUs assigned a weak association. Sensitivity and specificity with the test variable as the exposure were 0.4 and 0.75, respectively (Table 2). The model with the control variable as the exposure had an overall sensitivity of 0.4, and specificity of 0.76. 

As an alternative method of identifying OTUs associated with the response variable, we conducted Random Forest analysis. Each model, one with the test variable as the response, and one with the control variable, provides a score of importance of each predictor variable (OTUs + sex). We converted each score to a proportion out of 1 and set a cutoff of importance at 1/869 (the total number of predictors) to calculate sensitivity and specificity. The random forest model of the test variable was able to identify all of the strong (2) and medium (8) associated OTUs, and 3 of 10 weak OTUs, for an overall sensitivity of 0.65 and specificity of 0.7 (Table 2). The model with the control variable as the response identified 7 of the 20 OTUs associated with the simulated variable, and had a specificity of 0.35 and sensitivity of 0.63. 

### 3.7. Results of Real Data Demonstration

The study population was composed of 159 adolescent Chilean girls around age 15 with complete covariate data. There were 119 girls of normal weight and 40 girls who were overweight/obese. See Table 3 for further demographics and characteristics of the population.

Covariate adjusted WQS_RS_ without constraints in the positive or negative direction showed that the microbiome mixture in relation to BMI had a negative association, where 1228 out of 2000 of the estimated coefficients linking the WQS_RS_ mixture to BMI were negatively associated. We then ran a single adjusted WQS_RS_ with positive constraints (OR = 0.08, 95%CI = 0.001, 12.3), and a single adjusted WQS_RS_ with negative constraints (OR = 0.11, 95%CI = 0.002, 6.79) to confirm the negative direction before running the WQS_RSRH_. Although insignificant in both the negatively and positively constrained directions, the direction of the estimated odds ratios from both models indicated an overall negative association between the microbiome mixture and BMI group. The WQS_RSRH_ analysis was then performed with constraints in the negative direction. The WQS_RSRH_ analysis (Table 4) showed that the microbiome mixture had a negative association with BMI such that, for each unit increase in the WQS microbiome mixture, there was a 98% decrease in the odds of being overweight/obese versus normal weight (OR = 0.03, 95%CI: (0.00, 2.09). Of the 30 repeated holdout iterations, 28 (93%) had WQS estimates in the negative direction. This indicates that, as the abundance and/or the potency of the bacteria (with non-negligible weights) increase, the odds of being overweight/obese seems to decrease.

The genus-level threshold that informs which taxa have a greater impact than taxa assumed to be equally weighted can be found in Figure 4. Of 48 genera within the mixture, 7 were above the weight threshold, indicating that these genera had a greater contribution than all other genera to the negative association between the microbiome mixture and the odds of being overweight/obese in this population. These genera included *Bacteroides, Prevotella, Clostridium, Ruminococcus*, and unidentified genera from the Firmicutes, Actinobacteria, and Bacteroidetes phyla.

### 3.8. Comparison Methods Demonstration with Real Data

Adjusted linear regression analysis identified no association between BMI level and Shannon diversity (β = 0.0, 95%CI = −0.12–0.12) in the GOCS cohort. Adjusted PERMANOVA analysis showed a small but significant association between β-diversity and BMI level (R^2^ = 0.01, *p* = 0.002). Of the 109 ASVs that contributed to the highly weighted genera in the WQS_RSRH_ analysis, 37 (34%) were also selected by the Random Forest analysis, and 50 (46%) were also selected by SIMPER as associated with BMI level. *Bacteroides, Prevotella, Clostridium,* and *Ruminococcus* were identified in association with BMI across all three methods. *Collinsella*, *Shigella*, *Bifidobacterium*, *Akkermansia*, *Faecalibacterium*, *Lactobacillus*, *Lachnospira*, and *Robinsella* were identified by SIMPER and Random Forest, but not WQS_RSRH_.

## 4. Discussion

This simulation study demonstrated the novel use of the WQS analysis framework in microbiome data analysis. The WQS_RSRH_ model was able to detect a significant association in the correct direction between the test variable and the microbiome, in a dataset of 210 microbiome samples. With a WQS equi-weighted cut-point (1/868), average sensitivity and specificity across 30 random holdout models were 75% and 70%, respectively. In this simulation, we also demonstrated that the signal function based on the exp(t) improved specificity but was not different from less severe signal functions in assessing sensitivity. This method has potential for broad applications within microbiome research. WQS_RSRH_ can be used to assess associations between exposures of interest and the microbiome, as well as associations between the microbiome and health outcomes. Compared to standard methods of microbiome analysis, WQS_RSRH_ performed similarly or better than all other tested methods at identifying an overall association in the correct direction, and in sensitivity and specificity at correctly identifying the 20 OTUs with an association to the test variable. In further demonstration of the method with real data, the method was adjustable to accommodate the different composition of the dataset, including the use of ASV data instead of OTUs. Furthermore, the WQS_RSRH_ model found a negative association between the gut microbiome and BMI, and identified important bacterial taxa consistent with previously published studies. 

Our simulated variable was associated with the abundance of several OTUs across all participants. The abundance of those 20 OTUs contribute to the calculation of α-diversity, however, because α-diversity evaluates the association of single sample composition, and WQS_RSRH_ evaluates the OTU combination association across the population, it is not surprising that WQS_RSRH_ was able to detect an association with the test variable while α-diversity analysis was not. 

Alternatively, β-diversity directly compares composition of each sample to all others. There are many different methods to calculate similarity and dissimilarity distance for β-diversity analysis. In this analysis, we saw similar null results using both the Bray–Curtis and Aitchison distances. If the OTUs that we used to simulate an association were not major contributors to the overall composition of samples, PERMANOVA would not have found significant variance by the test variable. Moreover, if the 20 OTUs that were selected for association, were overwhelmed or drowned out by the richness and abundance of the other OTUs in each sample being compared, PERMANOVA would not have detected a significant amount of variance associated with the test variable. Likewise, SIMPER identifies which OTUs contributed the most to the variance detected between levels of exposure, so again if the OTUs with a simulated association were not major contributors to the composition of many of the samples, they would likely not have contributed much variance. It is noteworthy that SIMPER detected the same OTUs of the 20 simulated associations for both the test and control variables. It indicates that SIMPER is really constructed to identify the OTUs contributing most to the composition overall, those that are most abundant, and not necessarily the OTUs most associated with an exposure.

WQS_RSRH_ in contrast does not compare samples directly to each other, it evaluates the combination of all the OTUs across all samples, and weights the association of each OTU within the combination. This allows for identification of important OTUs even when their relative contribution to the composition of an individual sample may be small. It also allows for the identification of important OTUs across samples with very different composition. For instance, in observational studies, microbial composition of samples from different individuals can be difficult to compare to each other because there may be limited overlap in OTU composition, thus PERMANOVA and SIMPER analysis of β-diversity can fail to identify important associations. However, using WQS_RSRH_, OTUs are evaluated in combination across all samples, so two samples with completely different composition can both contribute heavily weighted OTUs to the combination, i.e., associated OTUs can be identified even when they are only in some of the study samples. WQS evaluates the association of the combination of OTUs, and indicates which are the most associated with exposure, without having to do direct sample comparisons, or relying only on the most abundant OTUs. The signal function in the weighted average across the random subsets further enhances the impact of random sets with important OTUs. 

The Random Forest analysis performed similarly to WQS_RSRH_ in sensitivity and specificity when using the simulated variable as the response. The test model also performed much better than the control model, indicating that it is better suited than methods like SIMPER to pick out the most associated OTUs. It is worth noting that when creating the simulated variable, all OTUs that were not assigned an association were assumed to have an association of 0. However, it is likely that some of those OTUs were correlated with some of the 20 OTUs that were assigned an association. These correlations likely account for some of the variation we see in sensitivity and specificity calculations across WQS_RSRH_, Random Forest, and SIMPER. 

While Random Forest performed well in this application, there are some potential advantages of using WQS_RSRH_ instead. WQS_RSRH_ simultaneously identifies the most important OTUs and estimates an overall mixture effect (or association in this case), instead of just identifying the importance of OTUs as the Random Forest does. In a situation like the one demonstrated in this simulation analysis, where there is an underlying association that is not detected by α and β-diversity, WQS_RSRH_ provides an additional measurement of association with the overall microbial composition that Random Forest does not. Additionally, incorporation and interpretation of covariates is simpler in WQS_RSRH_ models, as they are modeled as they would be in traditional regression methods instead of being included as a potential predictor along with the OTUs in a Random Forest model.

In the demonstration with data from GOCS, the WQS_RSRH_ method is adaptable to different datasets, and performed well in identifying bacteria related to high BMI. We were able to adjust the parameters of the model in several ways to accommodate the different datasets, and provide a demonstration of the WQS_RSRH_ method using ASV data as opposed to the OTU data that was used in the simulation study. We set the ranking mechanism to split the ASV abundance into three levels rather than four as shown in the simulation analysis. This adjustment was made to accommodate the smaller sample size and fewer microbiome mixture components (ASVs) in the GOCS dataset. The ranking levels can be adjusted to any number as appropriate for the dataset in use. Moreover, if the ranking is split once at zero, the microbiome mixture can be analyzed with presence/absence data rather than abundance. We also adjusted the number and size of the random subsets used to calculate the weighted index. The size of the random subsets should correspond to the number of observations in the dataset. The number of random subsets used is relatively arbitrary, but the larger the number of subsets, the more robust the estimate. We were also able to specify the data subsets to use in each repeated holdout iteration to ensure the variability of the categorical outcome in each subset. Although the WQS_RSRH_ estimate was not statistically significantly associated with BMI in this cohort, evidence of the trend in the negative direction was strengthened by 93% of the repeated holdout iterations producing a negative estimate. Analysis of association with α-diversity also found no association with BMI in this cohort. The bacterial taxa identified as heavily weighted within the negative association were consistent with bacterial genera negatively associated with obesity in other studies, and were also identified by the Random Forest and SIMPER methods. Although different species within the same genera may associate differently with obesity [40], several other studies have found a negative association between obesity and *Bacteroides* [41], *Ruminococcus* [42], and *Clostridium* [28] genera. While the Random Forest and SIMPER methods identified additional genera in association with BMI, it is important to note that these methods consider associations between each ASV and BMI individually, while WQS_RSRH_ is considering which ASVs are the most important in the gut microbiome mixture. 

There are many potential advantages of using WQS_RSRH_ for microbiome analysis, however, we are not suggesting that WQS_RSRH_ performs statistically differently than other methods that were compared. WQS_RSRH_ works with both continuous and categorical variables, and allows for the adjustment of covariates in an interpretable fashion. It accounts for the correlated nature of the taxa within the microbiome, and gives an overall effect estimate and the weight of importance when all taxa in the index are considered together. WQS_RSRH_ allows for analysis of associations in positive and negative directions separately, and allows flexibility in choosing the signal function in the weighting step. It accommodates samples from populations with widely varied microbial composition, identifying associations with OTUs present in a relatively small proportion of the population. WQS_RSRH_ also gives robust estimates over many repetitions of the analysis. WQS_RSRH_ could be used in a broad range of health research, as well as in a drug discovery framework. It could identify groups of bacteria that are associated together with an outcome of interest, which could be targeted together in developing probiotics. When the analyst is interested in determining a small subset of OTUs associated with the outcome, as in a drug discovery framework, the bottom 90–95% of the weights can be set to zero to test for significance in the top 5–10%. This may lead to better identification of bacteria to include in probiotics that will be successful within the gut microbiome ecosystem.

While this demonstration of the application of WQS to microbiome data establishes a novel analytical method with potential for broad use, it does have some limitations. The first is that, while the overall WQS_RSRH_ estimate identifies the direction of association between the variable of interest and the abundance of taxa within the microbiome, it is not directly translatable to a value measure (i.e., a measureable unit of some health outcome). However, the current standard microbiome analyses using α-diversity, β-diversity, and principal components analysis suffer from a similar limitation. Another limiting consideration is that the sensitivity and specificity of this WQS_RSRH_ method application depends on the number of taxa in the data set, and the cutoff point chosen to identify the taxa of most importance. Of course, in an analysis of microbiome data with a real variable with unknown association, we would not know the truth to determine a good cutoff for both sensitivity and specificity. As demonstrated with the GOCS data, the equi-weighted cut point can be used as a default with unknown sensitivity and specificity. However, relevant cut points could be determined by investigating the significance of the index across repeated holdout data sets. The use of 16S rRNA sequencing with taxonomic assignment down to the genus level is also somewhat controversial among microbiome researchers. However, because the primary purpose of this analysis was to demonstrate the analytical method, the choice of grouping results at the genus level rather than the family level is not a major limitation. Lastly, in calculating the α and β-diversity for comparison to WQS_RSRH_, we used the same data set, which was limited to OTUs detected in at least 10% of samples, and converted to relative abundance. These processing steps affected the diversity calculations that rely on singletons. While this allows us to do a direct comparison with the WQS_RSRH_ method, these calculated values are not generalizable outside this study.

This simulation study is the first step in exploring the use of WQS methods on the analysis of microbiome data. We plan to apply WQS methods to other forms of microbiome data beyond 16s rRNA amplicon sequencing, including metagenomic sequence data. Further development of the general WQS method is still underway, including the use of stratification, and Bayesian statistical applications, thus as those methods continue to develop, we will test their application on microbiome data. 

## 5. Conclusions

This study demonstrated the application of WQS_RSRH_ to microbiome sequencing abundance data. In our analysis, the WQS_RSRH_ method was able to detect a significant association between the test variable and the overall abundance of the microbiome in the correct direction, and identified the assigned associated OTUs with acceptable levels of sensitivity and specificity. In further demonstration with real data, the model identified a direction of association with heavily weighted taxa consistent with other studies of BMI and the gut microbiome. This method has potential for broad application within microbiome research, and we plan to continue to refine and apply WQS methods to different analyses of microbiome data. 

## Figures and Tables

**Figure 1 ijerph-20-00094-f001:**
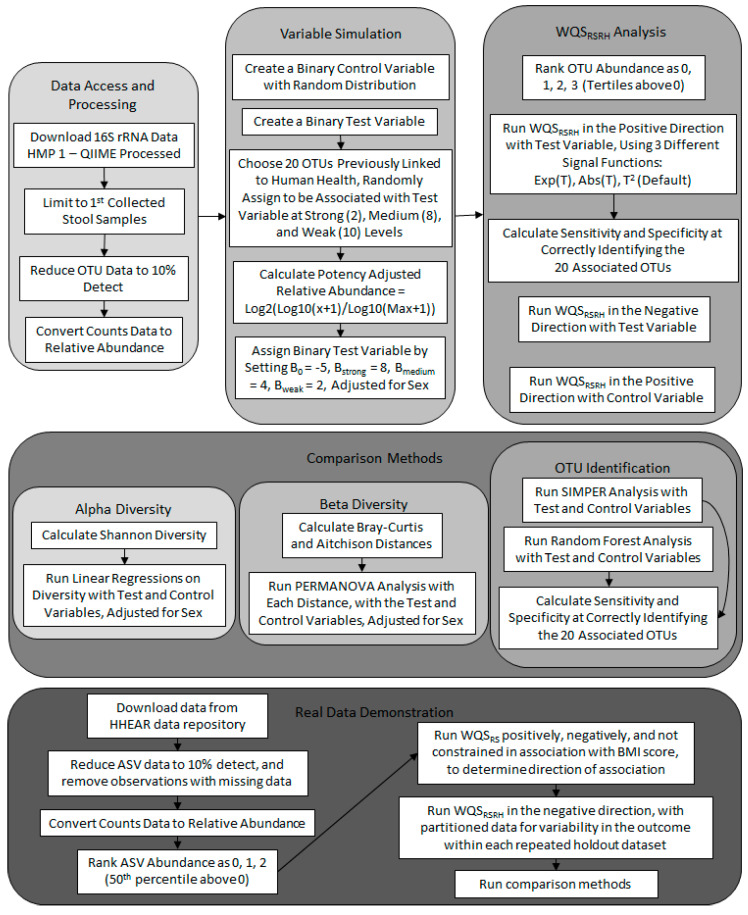
Study Schematic. A simplified flow-chart of the study procedures. Abbreviations: Amplicon Sequence Variant (ASV); Body Mass Index (BMI); Human Health Exposure Analysis Resource (HHEAR); Human Microbiome Project (HMP); Operational Taxonomic Unit (OTU); Permutational Analysis of Variance (PERMANOVA); Quantitative Insights Into Microbial Ecology (QIIME); Similarity Percentage (SIMPER); Weighted Quantile Sum Regression with Random Subsets (WQS_RS_); Weighted Quantile Sum Regression with Random Subsets and Repeated Holdouts (WQS_RSRH_).

**Figure 2 ijerph-20-00094-f002:**
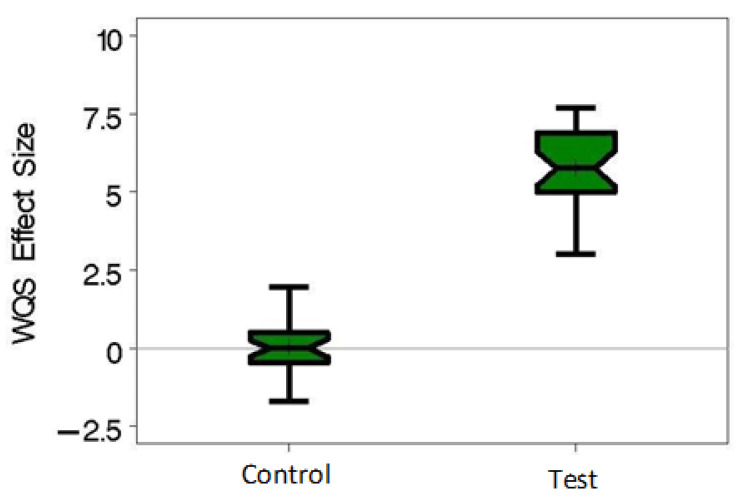
Simulated Variable Associations with the Human Gut Microbiome. Box plots show the beta coefficients estimated as the association between the human gut microbiome and the simulated test and control variables, using weighted quantile sum regression with random subsets and 30 repeated holdouts. Associations with error bars that do not cross 0 are considered statistically significant. Data come from the Human Microbiome Project I with simulated test and control variables. Abbreviations: Weighted Quantile Sum Regression (WQS).

**Figure 3 ijerph-20-00094-f003:**
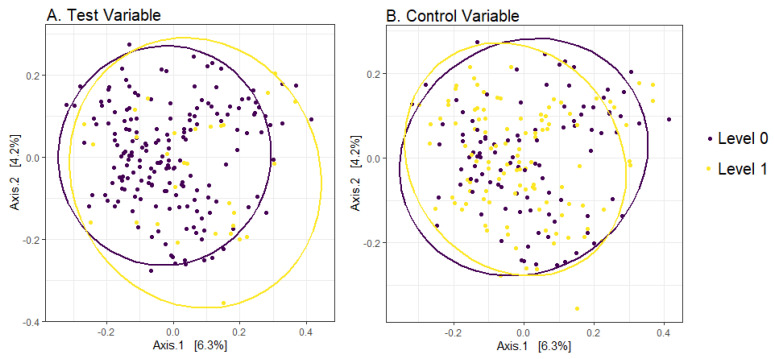
Human Gut Microbiome Beta Diversity by Level of the Simulated Variables. Bray–Curtis dissimilarity distance (beta diversity) shown using multidimensional scaling (MDS) ordination plots of (**A**) the test variable, and (**B**) the control variable. Data points represent individual observations. Data points closer together represent gut microbiome composition that is more similar, while data points farther apart represent gut microbiome composition that is more different. Data come from the Human Microbiome Project I with simulated test and control variables.

**Figure 4 ijerph-20-00094-f004:**
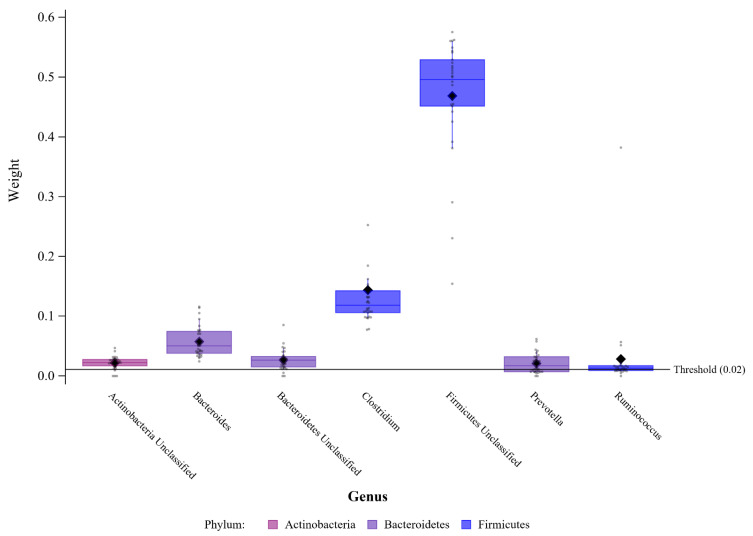
Bacterial Genera Negatively Associated with Obesity. Data points indicate the sum of weights in association with BMI level for each of the 30 repeated holdout analyses (from weighted quantile sum regression with random subsets and 30 repeated holdouts) for amplicon sequence variants (ASVs), pooled by genus and sorted by phylum. Only genera with pooled weights above the equi-weighted threshold are shown, and are considered important within the gut microbiome mixture. Box plots show 25th, 50th, and 75th percentiles of the sum of weights within genera. Closed diamonds show the sum of the mean weights within each genus. Data come from the Growth and Obesity Cohort Study.

**Table 1 ijerph-20-00094-t001:** Description of the 20 operational taxonomic units (OTUs) that were assigned an association with the simulated binary (test) variable. In the weighted quantile sum regression, relative abundance of each OTU within the gut microbiome from each participant was ranked as 0 if relative abundance was 0, and then by tertiles above 0 for ranks 1–3. Table columns represent the taxonomy, sample size, row percent of observations at each rank level, median and maximum of non-zero relative abundance, and level of assigned association with the test variable (β = 8, 4, 2 for strong, medium, and weak, respectively) for the 20 OTUs assigned an association.

Phylum	Family	Genus	N	Rank 0 %	Rank 1 %	Rank 2 %	Rank 3 %	Median of Non-Zero Values ^1^	Maximum ^1^	Assigned Association
Actinobacteria	Micrococcaceae	*Rothia* [23]	210	81.4	6.2	6.2	6.2	0.12	8.33	Medium
	Coriobacteriaceae	*Atopobium* [23]	210	82.9	5.7	5.7	5.7	0.03	0.94	Weak
		*unclassified* [23]	210	83.3	5.7	5.2	5.7	0.05	1.65	Medium
Bacteroidetes	Bacteroidaceae	*Bacteroides* [24]	210	82.4	5.7	6.2	5.7	0.03	0.75	Strong
	Rikenellaceae	*Allistipes* [1]	210	87.1	4.3	4.3	4.3	0.37	16.06	Medium
Firmicutes	Staphylococcaceae	*Staphylococcus* [25]	210	35.2	21.4	21.9	21.4	0.51	4.71	Weak
	Lactobacillaceae	*Lactobacillus* [26,27]	210	85.7	4.8	4.8	4.8	0.03	0.88	Weak
	Eubacteriaceae	*Eubacterium* [24,28]	210	63.8	11.9	12.4	11.9	0.24	3.79	Medium
	Lachnospiraceae	*Coprococcus* [29,30]	210	86.7	4.3	4.8	4.3	0.05	2.03	Weak
		*Dorea* [29,30]	210	58.1	13.8	14.3	13.8	0.81	19.75	Medium
		*Roseburia* [1,28]	210	57.6	14.3	13.8	14.3	0.19	3.25	Weak
		*unclassified* [29,30]	210	58.6	13.8	13.8	13.8	0.28	14.60	Strong
	Ruminococcaceae	*Faecalibacterium* [1,31]	210	61.9	12.9	12.4	12.9	0.22	4.90	Medium
		*unclassified* [25]	210	85.2	4.8	5.2	4.8	0.07	2.80	Weak
	Erysipelotrichaceae	*Coprobacillus* [23]	210	81.0	6.2	6.7	6.2	0.05	1.24	Medium
		*Holdemania* [23]	210	87.6	4.3	3.8	4.3	0.05	1.97	Weak
		*Solobacterium* [23]	210	86.2	4.8	4.3	4.8	0.02	0.83	Weak
		*Turicibacter* [23]	210	84.8	5.2	4.8	5.2	0.04	1.24	Medium
Proteobacteria	Enterobacteriaceae	*Serratia* [32]	210	80.0	6.7	6.7	6.7	0.03	0.84	Weak
Verrucomicrobia	Verrucomicrobiaceae	*Akkermansia* [25]	210	88.1	3.8	4.3	3.8	0.03	1.65	Weak

^1^ Shown in percent relative abundance.

**Table 2 ijerph-20-00094-t002:** Sensitivity and specificity in identifying the 20 associated operational taxonomic units (OTUs) for weighted quantile sum regression with random subsets and repeated holdouts (WQS_RSRH_), similarity percentage (SIMPER), and Random Forest models. For WQS_RSRH_ the sensitivity and specificity were averaged across 30 repeated holdout datasets with a cutoff of 0.00115 (=1/868). For SIMPER, one model was conducted with 999 permutations, and the cutoff of importance was 70% cumulative variance. For Random Forest, one model was conducted with 100 trees, and importance scores were converted to a proportion, with a cutoff of 0.00115.

	Denominator	WQS_RSRH_ Proportion ^1^	SIMPER Proportion ^1^	Random Forest Proportion ^1^
Overall Sensitivity	20	0.75 (0.60–0.90)	0.40 (0.19–0.61)	0.65 (0.44–0.86)
Overall Specificity	848	0.70 (0.54–0.86)	0.75 (0.72–0.78)	0.70 (0.67–0.73)
Sensitivity: Strong	2	0.87	0.50	1.00
Sensitivity: Medium	8	0.90	0.63	1.00
Sensitivity: Weak	10	0.61	0.20	0.30

^1^ Confidence intervals (CIs) were calculated as *p ± *1.96** √pq/n*, where *p* is the proportion estimate, *q* is *1-p,* and *n* is the number of observations. For the WQS_RSRH_ CI, *n* = 30 for the number of repeated holdouts averaged in the estimate, and for SIMPER and Random Forest CIs, *n* = the value from the denominator column. CIs are included for the overall estimates of specificity and sensitivity, but not for the subset sensitivity analyses due to small denominators. CIs are provided to demonstrate the reasonable range of the estimate, not to indicate statistical significance between the methods.

**Table 3 ijerph-20-00094-t003:** Description of the study population from the Growth and Obesity Cohort Study data used in the illustration analysis.

Characteristic	N	Normal Weight, N = 119 ^1^	Overweight/Obese, N = 40 ^1^	*p*-Value ^2^
Age (years)	159	15.4 (0.6)	15.3 (0.6)	0.2
Number of days breastfed as infant	159	89.8 (76.4)	104.4 (87.8)	0.4
Birth mode (vaginal versus C-section)	159			0.2
C-section		31 (26%)	15 (38%)	
Vaginal		88 (74%)	25 (62%)	
Were antibiotics used in the past 6 months (yes vs. no)	159	19 (16%)	5 (12%)	0.6
Maternal education	159			0.12
High school or less		94 (79%)	36 (90%)	
More than high school		25 (21%)	4 (10%)	

^1^ Mean (SD); n (%) ^2^ Welch Two Sample *t*-test or Pearson’s Chi-squared test.

**Table 4 ijerph-20-00094-t004:** Estimates of association with body mass index (BMI), shown as odds ratios from the Weighted Quantile Sum regression with Random Subsets and Repeated Holdouts analysis. The weighted quantile sum (WQS) variable represents the estimate for the association between the gut microbiome mixture and BMI. Odds ratios with confidence intervals (CIs) that do not cross 1.0 are considered statistically significant. Data come from the Growth and Obesity Cohort Study.

Variable	OR (95% CI)
WQS	0.03 (0.00, 2.09)
Maternal Education (More than High School)	0.22 (0.00, 89.77)
Birth Mode (Vaginal)	0.72 (0.34, 1.52)
Age at Stool Sample	0.68 (0.37, 1.25)
Number of Days Breastfed as an Infant	1.00 (1.00, 1.01)
Antibiotic Use in Past 6 Months	0.50 (0.00, 3.08)

## Data Availability

Data come from The Human Microbiome Project I, the Growth and Obesity Cohort Study, and the Human Health Exposure Analysis Resource, and are publicly available at https://www.hmpdacc.org/hmp/, and DOIs doi.org\\10.36043/1977_480 and doi.org\\10.36043/1977_490. Code used in this analysis is available at github.com/ShoshannahE/WQS-Microbiome (DOI: 10.5281/zenodo.7017101).

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
