# Peer review of "Human Microbiome Mixture Analysis Using Weighted Quantile Sum Regression"

_ijerph, 2022, doi:10.3390/ijerph20010094_

Round 1

Reviewer 1 Report

Eggers et al., did an interesting study on human microbiome mixtures analysis in both simulated data from HMP1 and real data of a cohort. The authors have done comprehensive analyses and the manuscript is well-written. The manuscript is of interest to the readers of the International Journal of Environmental Research and Public Health. Please see my comments below:

1.      Line 129-130, the selection of 20 OTUs are relatively small given the total of 868 OTUs. The authors hand-selected those OTUs according to literatures. I am wondering why 20, is that because only 20 OTUs were mentioned in the literature OR the ability of WQS of taking too many OTUs as a mixture is limited? How about let machine learning models select important OTUs, for example, the authors used random forest, feature importance methods can be used to select top OTUs. Will that have a better result than hand-selected features? The authors may want to clarify the rationale a little more.

2.      Line 148, Table1. Consistent font.

3.      Line 148, I hold my conservative idea that the amplicon sequencing method is not accurate when goes to the genus level. The most accurate for amplicon sequencing can go as deep as the family level. There are other streams of thoughts that they can go deep to genus level. So either way is fine.

4.      Line 250, the HMP amplicon data is in OTUs, the HHEAR data is in ASVs. OTUs cluster 97% of similarity while ASV is the exact sequence. There is a big debate in the field about whether to use OTUs or ASVs. Why do the authors use two different methods of representing microbiomes in the manuscript?

Author Response

Thank you for your thoughtful comments. Please see a point by point response below.

Reviewer 1 Comments

 Response

1.      Line 129-130, the selection of 20 OTUs are relatively small given the total of 868 OTUs. The authors hand-selected those OTUs according to literatures. I am wondering why 20, is that because only 20 OTUs were mentioned in the literature OR the ability of WQS of taking too many OTUs as a mixture is limited? How about let machine learning models select important OTUs, for example, the authors used random forest, feature importance methods can be used to select top OTUs. Will that have a better result than hand-selected features? The authors may want to clarify the rationale a little more.

The number 20 was chosen somewhat arbitrarily because it allowed for an increasing number of OTUs to be assigned increasing strength of associations with the simulated outcome, while keeping the simulation step relatively simple. It is not a reflection of the power of the WQSRSRH model to detect heavily weighted mixture components. Ultimately, any number we would have chosen for this step would have been somewhat arbitrary. Methods like random forest models could not have been used to select for important OTUs in this simulation step because we did not yet have an outcome variable for the OTUs to be associated with. Additional explanation has been added on lines 135-136.

2.      Line 148, Table1. Consistent font.

All tables have been updated with consistent formatting.

3.      Line 148, I hold my conservative idea that the amplicon sequencing method is not accurate when goes to the genus level. The most accurate for amplicon sequencing can go as deep as the family level. There are other streams of thoughts that they can go deep to genus level. So either way is fine.

Thank you for bringing up this point. Considerations around the use of 16s data with taxonomic assignment at the genus level has been added to the discussion (L. 599-603).

4.      Line 250, the HMP amplicon data is in OTUs, the HHEAR data is in ASVs. OTUs cluster 97% of similarity while ASV is the exact sequence. There is a big debate in the field about whether to use OTUs or ASVs. Why do the authors use two different methods of representing microbiomes in the manuscript?

Both of these data sets are publicly available in pre-processed abundance tables. These two data sets happened to be processed differently. We decided to keep the data in these previously processed formats as a way to demonstrate that the WQSRSRH model is versatile and will work with data processed using different pipelines, e.g. OTUs and ASVs. Additional explanation of this was added to the discussion (L. 477-478, 541-542).

Reviewer 2 Report

Recently, there has been a rapid acquisition of big data on the human microbiome. It is quite obvious that the tools for bioinformatics processing of these data must be developed and improved. The authors present the results of applying random subset weighted quantile sum regression with repeated holdouts (WQSRSRH) to search for relationships between predictors and ‘omic data. The application of WQSRSRH to the microbiome proved successful and allowed the authors to identify an inverse association between body mass index and the gut microbes.

I have no comments on the structure and content of the manuscript, my comments are of a technical nature:

1.       In the Section “3.8. Comparison Methods Demonstration with Real Data” authors identified four genera as important genera in the negative association with BMI. These are Bacteroides, Prevotella, Clostridium, and Ruminococcus, in the Abstract, they mentioned only three of them, Bacteroides, Clostridium, and Ruminococcus. I could not find a reason to exclude the Prevotella genus from this explanation.

2.       Line 432-433: the correct names of bacteria phyla are Actinobacteria, Firmicutes

3.       Fig. 4 and Legend to Fig. 4 – the same corrections

4.       The list of references must be corrected and checked (exp. ref. 25)

Author Response

Thank you for your thoughtful review. Please see a point by point response below:

Reviewer 2 Comments

 Response

1.       In the Section “3.8. Comparison Methods Demonstration with Real Data” authors identified four genera as important genera in the negative association with BMI. These are Bacteroides, Prevotella, Clostridium, and Ruminococcus, in the Abstract, they mentioned only three of them, Bacteroides, Clostridium, and Ruminococcus. I could not find a reason to exclude the Prevotella genus from this explanation.

Prevotella has been added to the abstract as it was unintentionally excluded in the previous version.

2.       Line 432-433: the correct names of bacteria phyla are Actinobacteria, Firmicutes

These typos have been corrected.

3.       Fig. 4 and Legend to Fig. 4 – the same corrections

These typos have been corrected.

4.       The list of references must be corrected and checked (exp. ref. 25)

All references have been updated and reformatted.